# Altitudinal Gradient and Soil Depth as Sources of Variations in Fungal Communities Revealed by Culture-Dependent and Culture-Independent Methods in the Negev Desert, Israel

**DOI:** 10.3390/microorganisms11071761

**Published:** 2023-07-05

**Authors:** Isabella Grishkan, Giora J. Kidron, Natalia Rodriguez-Berbel, Isabel Miralles, Raúl Ortega

**Affiliations:** 1Institute of Evolution, University of Haifa, 199 Aba Khoushy Ave, Mount Carmel, Haifa 3498838, Israel; 2Institute of Earth Sciences, The Hebrew University of Jerusalem, Givat Ram Campus, Jerusalem 91904, Israel; kidron@mail.huji.ac.il; 3Department of Agronomy & Center for Intensive Mediterranean Agrosystems and Agrifood, University of Almeria, E-04120 Almería, Spain; nrodfer@ual.es (N.R.-B.); imiralles@ual.es (I.M.); rortega@ual.es (R.O.)

**Keywords:** culturable communities, DNA-based communities, fungal diversity, melanin-containing fungi, water availability

## Abstract

We examined fungal communities in soil profiles of 0–10 cm depth along the altitudinal gradient of 250–530–990 m.a.s.l. at the Central Negev Desert, Israel, which benefit from similar annual precipitation (95 mm). In the soil samples collected in the summer of 2020, a mycobiota accounting for 169 species was revealed by both culture-dependent and culture-independent (DNA-based) methodologies. The impact of soil depth on the variations in fungal communities was stronger than the impact of altitude. Both methodologies displayed a similar tendency in the composition of fungal communities: the prevalence of melanin-containing species with many-celled large spores (mainly *Alternaria* spp.) in the uppermost layers and the depth-wise increase in the proportion of light-colored species producing a high amount of small one-celled spores. The culturable and the DNA-based fungal communities had only 13 species in common. The differences were attributed to the pros and cons of each method. Nevertheless, despite the drawbacks, the employment of both methodologies has an advantage in providing a more comprehensive picture of fungal diversity in soils.

## 1. Introduction

It is now well established that the formation, development, and functioning of soil microbial communities are faced with strong environmental selection at different geographic scales (e.g., [1,2,3]). For that reason, research on the diversity and distribution of soil fungi in stressful habitats is of utmost importance, as it may shed light on the mechanisms of survival and adaptation of microorganisms in extreme environments. Deserts represent a clear example of such stressful habitats where severe climate and limited resources greatly influence the biota formation and functioning [4].

Deserts in Israel constitute more than 60% of its territory [5] covering the semi-arid to extremely arid regions with annual rainfall from 300 mm to 25 mm, respectively. The gradient also includes a range of altitude and vegetation types, as well as a variety of microclimatic and edaphic conditions. Such a combination of ecological factors offers an excellent opportunity to study the diversity of desert soil fungi and their distribution pattern on a broad environmental scale.

Our previous mycological studies were conducted in different locations at the central and southern Negev Desert [6,7,8,9,10,11] and at the Arava Valley along the eastern limit of the Negev [12]. These studies have shown that in spite of a hostile environment (high solar radiation, very high summer temperatures, low water availability, and oligotrophic conditions), the soils of the Israeli deserts maintained rich cultivable mycobiota, accounting for nearly 430 identified species from 138 genera. The mycobiota displays remarkable adaptive strategies (i.e., melanin-containing fungi with large, thick-walled, and multicellular conidia dominated the majority of topsoil communities; thermophilic *Aspergillus fumigatus* prevailing in the extremely hot localities; relatively small spatiotemporal variations in the community composition) aiming to cope with the harsh climatic and edaphic arid conditions [13].

In our previous studies, we mainly focused on the dynamics of soil microfungal communities along latitudinal [6,9,12] or microclimatic [8] gradients. On a microscale, the differences between sun-exposed and under-canopy communities were also examined [7,9,11,12], as well as the variation in fungal communities through soil depth [10,11].

The effect of altitude on free-living soil fungal communities was not extensively studied all over the world. In temperate regions, the increase in fungal abundance and diversity along the elevation gradient has been revealed in some studies [14,15], while no consistent pattern was found in other cases [16,17,18]. To the best of our knowledge, the impact of altitude on soil fungal communities in desert regions was not previously investigated.

The main goal of the current research addresses the combined effect of altitude and soil depth on the divergence of fungal communities in the Central Negev. For that purposes, three sampling sites located at different elevations but receiving similar annual precipitation were chosen: Nahal Nizzana at 250 m.a.s.l, Nahal Boker at 530 m.a.s.l, and Har Harif at 990 m.a.s.l. Along the chosen altitudinal gradient, the effect of dew and fog on the communities of saxicolous lichens has been investigated [19]. The altitudinal variability in cobble-inhabited lichens and cyanobacteria in the Negev desert was attributed to the increase in dew with elevation [19,20]. At the same time, recent research has shown the inability of dew and fog to wet the soil surface (e.g., [21]) and thus to affect soil moisture content, which directly influences the development of fungal communities.

Along the altitudinal gradient, we examined the composition, structure, diversity level, and quantitative characteristics (isolate density) of fungal communities at three soil depths. We hypothesized, based on our previous findings, that soil depth would be a stronger driver of variations in the community characteristics than altitude because of sharper changes in the edaphic conditions with depth (increased salinity, limited water infiltration). Importantly, we surveyed the soil fungal communities by both culture-dependent and culture-independent methodologies, aiming to obtain a broader community composition.

## 2. Materials and Methods

### 2.1. Site Description

Three research sites were chosen in the Central Negev Desert: Nahal Nizzana (NN) at 250 m above sea level (34°23′ E, 30°58′ N), Nahal Boker (NB) at 550 m.a.s.l (34°46′ E, 30°56′ N), and Har Harif (HH) at 990 m.a.s.l (34°37′ E, 30°29′ N). The research sites are shown in Figure 1.

The climate of the area is arid. Long-term average annual rain precipitation is similar in all sites, ~95 mm, with rains falling principally between November and April [22]. Dew and fog exhibited a clear altitudinal increase with an average daily amount being 0.1, 0.2, and 0.3 mm in NN, NB, and HH, respectively [23]. Air temperatures decrease with the increase in altitude, with mean annual temperatures of 20.0 °C, 17.9 °C, and 17.3 °C; maximum daily temperatures averaging 26.5, 24.7, and 23.8 °C during the hottest month (July), and minimum temperatures averaging 11.8, 9.3, and 9.0 °C during the coldest month (January) at NN, NB, and HH, respectively [24]. Annual evaporation shows the same tendency with ≈2700 mm in NN [24], ≈2600 mm in NB [25], and ≈2500 mm in HH [26].

The main lithology of all sites consists of limestone. While eroded alluvial fans are distributed at NN, limestone hills with patches of loessial soils (with fines content, i.e., silt and clay of 60–70% and pH of 8.0–8.5) characterize NB and HH. The surface of the sites is covered by a thin (~1 mm) cyanobacterial biocrust (biological soil crust, BSC) with *Microcoleus vaginatus* predominating (B. Bűdel, personal communication); NN and NB are also characterized by high stoniness, with 90%, 50%, and 20% cover of pebbles and cobbles at NN, NB, and HH, respectively. Vegetation is sparse, consisting mainly of semi-shrubs and dwarf shrubs, covering approximately 5% in NN, 10–15% in NB, and 15–20% in HH. The predominant shrubs are *Zygophyllum dumosum* Boiss. (NN), *Z. dumosum* and *Artemisia herba-alba* Asso (NB), and *A. herba-alba* (HH).

### 2.2. Sampling Design

The sampling was conducted in July, 2020 at three sites along the altitudinal gradient (Figure 1). The samples were collected from the soil layers of 0–1 cm (which include the biological soil crust), 1–5 cm, and 5–10 cm in four replicates (four randomly chosen vertical profiles, 20–30 m apart from each other at a hilltop of each site). During sampling, a ruler was inserted into profiles and the horizontal cuts were made at the height of 1, 5, and 10 cm. The samples were collected by means of a wide spatula into sterile paper bags. Altogether, from all soil profiles, 36 soil samples were collected and processed for the isolation of culturable fungal communities. Fungal DNA extraction was performed in the samples mixed from four subsamples collected in the aforementioned four vertical profiles from two soil layers—0–1 and 5–10 cm, and altogether, six soil samples were processed for the fungal DNA extraction. All samples were sieved through a 2 mm sieve in order to remove rock particles and debris and were then stored in sterile paper bags in freeze conditions in the lab prior to subsequent analyses.

### 2.3. Characterization of Fungal Communities

#### 2.3.1. Culture-Dependent Approach

For isolation of fungi, the soil dilution plate method [27] was employed. Ten grams of soil samples were used in a dilution series. Streptomycine (Spectrum Chemical Mfg. Corp, Gardens, NY, USA) was added to each medium (100 µg/mL) to suppress bacterial growth. Soil suspension in an amount of 0.5–1 mL from the dilutions 1:10 and 1:100 (soil:sterile water) was mixed with an agar medium at 40 °C in Petri dishes (90 mm diameter). Two culture media with different C and N sources were employed: Malt Extract Agar (MEA) and Czapek’s Agar (CzA) (Sigma-Aldrich Inc., St. Louis, MN, USA). The plates were incubated at 25 °C in darkness for 10 to 15 days (3 plates for each medium).

After incubation, the emerging fungal colonies were transferred to MEA and CzA for purification and further taxonomic identification. In an attempt to induce sporulation, all non-sporulating isolates were also grown on Oatmeal Agar (Sigma-Aldrich Inc., St. Louis, USA) as recommended by Bills et al. [28], and on Water Agar (agar—20 g, water—1000 mL). Taxonomic identification was mainly based on morphological characteristics of fungal isolates. The three most frequently occurring non-sporulating strains were subjected to molecular identification performed at Hy Laboratories Ltd., Rehovot, Israel. The procedure is described in detail in our previous publications (e.g., [11]). All names of the identified species are cited according to the Species Fungorum database (www.speciesfungorum.org, assessed on 28 April 2023).

#### 2.3.2. Culture-Independent Approach Based on Next-Generation Sequencing (NGS)

Fungal DNA was extracted from 0.3 g of a composite soil sample taken from the four profiles using a PowerSoil DNA Isolation kit (QIAGEN Inc., Hilden, Germany), following the manufacturer’s instructions. A ND-2000 Nanodrop spectrophotometer (Thermo Fisher Scientific, Waltham, MA, USA) was used to quantify the DNA concentration (ng/μL). To characterize fungal community composition, fungal ITS genes were amplified and sequenced by polymerase chain reaction (PCR) using the primer pairs ITS86F/ITS4 [29]. To verify no contamination throughout the DNA extraction, one blank control was added using one of the kit tubes. Negative controls were also tested (1 for every 96-well plate = 4 perMiSeq run), checked to be clean (no bands present), and still sequenced on the MiSeq to show that no substantial reads were generated on this barcode combination. The final sequence files were then processed using QIIME2 software (version 19.10) [30] following the protocol on the Microbiome Helper website (Amplicon SOP v2 (qiime2 2019.7)). Taxonomy assignments of fungal phylotypes (ASVs—Amplicon Sequence Variants) were performed against the UNITE database (version 7) (https://unite.ut.ee/, assessed on 28 April 2023).

### 2.4. Data Analyses

The density of fungal isolates was expressed as “colony forming units” (CFU) per gram of dry soil. The relative abundance of species was calculated as a number of isolates of a particular species in the sample/total number of all isolates in the sample. Similarly, the relative abundance of fungal species detected by NGS was calculated as the number of sequences of a particular species/total number of sequences read in each sample. The diversity analysis of both culturable and DNA-based communities were based on the Shannon–Wiener (H) and evenness (J = H/Hmax) indices [31].

To analyze the variations in fungal community structure, two major groupings were chosen: (i) the core desert group of melanin-containing fungi resistant to different kinds of abiotic and biotic stresses; in this group, species with large multicellular spores were examined separately, and (ii) the group of fungi with opposite traits, that is, light-colored species which produce small one-celled spores, able to penetrate deeper soil layers. The contribution of each group to the communities was estimated as the sum of relative abundances of species comprising the group.

Statistical analysis was conducted using XLSTAT (http://www.xlstat.com, assessed on 28 April 2023). A two-way unbalanced ANOVA with interactions was used to test the effect of altitude and soil depth, separately and in interaction, on diversity characteristics, abundance of fungal groups, and isolate density. To evaluate the similarity between communities, the clustering of communities based on species relative abundances was performed by the unweighted pair-group average method with the Bray–Curtis distance as the distance coefficient.

## 3. Results

### 3.1. Culturable Fungal Communities

#### 3.1.1. Composition and Diversity

Altogether, 70 identified species were isolated from the phyla Mortierellomycota (1 species), Ascomycota (68), and Basidiomycota (1) (Table 1). The species belong to 48 genera with *Alternaria*, *Chaetomium* (7 species each), and *Fusarium* (4) being the most numerous. Seven types of fungal strains, which remained non-sporulating in culture, have not been identified.

From the NN, NB, and HH soil profiles, 46, 42, and 43 species, were respectively isolated. In each soil layer, the HH and NN fungal communities were the most and least diverse, respectively (Table 2a). The biocrust communities in each site were characterized by the highest species richness, heterogeneity, and evenness. At NN, the decrease in diversity level with depth was the most pronounced, with the extremely low Shannon and evenness indexes in the 1–5 cm layer (Table 2a).

In all topsoil crust communities, melanin-containing species prevailed, with relative abundance of 71.2–89.7% (Figure 2). In turn, species with large (>20 µm) multicellular spores (mostly *Alternaria atra*, *A. alternata*, *A. chlamydosporigena*, and *Stemphylium botryosum*—Table 1) comprised 41.2–60.8% of all isolates. The abundance of melanin-containing species substantially decreased from the surface to the 5–10 cm layers (Figure 2)—both overall (4.4, 3.6, and 3.5 in the NN, NB, and NN profiles, respectively) and more sharply—for species with many-celled spores (4.9, 13.4, and 14.2-fold in the NN, NB, and HH profiles, respectively).

Simultaneously, the abundance of light-colored species producing small one-celled spores remarkably increased in the deeper soil layers (10.1, 5.3, and 2.7, 5.3, 10.1-fold in the NN, NB, and HH profiles, respectively) peaking in the 1–5 cm layer at NN—83.2% (Figure 2). The following species from this group were dominant or frequently occurring at the depth of 1–10 cm: *Penicillium simplicissimum* (HH and NB), *Talaromyces variabilis* (HH), *Gymnoascus reesii* (NB), and *Parengyodontium album* (overwhelmingly prevailing at NN; Table 1).

Cluster analysis based on species relative abundances separated the fungal communities into three distinct groups (Figure 3). The first group consists of the biocrust communities (right side of Figure 3) due to the similarity of the dominant and frequent melanin-containing species. The second group (left side of Figure 3) includes the communities from the 1 to 10 cm layers of the HH and NB soil profiles dominated and co-dominated by *P. simplicissimum.* The communities from the 1 to 10 cm layers at NN formed a distinct group (located in the center of Figure 3) due to the overwhelming prevalence of *P. album* (Table 1).

#### 3.1.2. Density of Microfungal Isolates

In the studied profiles, the highest isolate densities were found in the topsoil layers (Figure 4). At HH and NB, CFU numbers markedly decreased at the depth of 1–5 cm and then increased (due to *P. simplicissimum*) in the 5–10 cm layers, almost reaching the topsoil values. At NN, the density of microfungal isolates was 1.6–5.9-fold lower than in the corresponding soil layers at HH and NB, and remarkably decreased with depth.

### 3.2. Effect of Altitude and Soil Depth on the Characteristics of Microfungal Communities

The two-way unbalanced ANOVA test showed that the altitude, soil depth, and their interactions explain 46–92% of the variability in the community characteristics (the R^2^ column in Table 3). Soil depth most strongly and highly significantly influenced all community parameters, except for the density of microfungal isolates (Table 3). Altitude also strongly affected the variations in community characteristics except for the abundance of melanin-containing species with large multicellular spores and species richness. Some parameters—heterogeneity, evenness, and the abundance of melanized species with multicellular spores, were significantly sensitive to the cumulative effect of altitude and soil depth (Table 3).

### 3.3. Fungal Communities Revealed by DNA Extraction

#### Composition and Diversity

Altogether, ASVs belonging to 113 free-living fungal species were obtained, 68 affiliated to the species or genus level, 8—to the order level, 1—to the class level, and 18—to the phylum level (Table 4). ASVs belonged to five phyla: Ascomycota (94 species), Basidiomycota (13), Chytridiomycota (4), Glomeromycota (2), and Mortierellomycota (1). *Alternaria, Chaetomium*, and *Cladosporium* were the most numerous genera (6, 5, and 4 species, respectively). Thirteen species (*Acremonium alternatum, Alternaria chlamydospora, A. chlamydosporigena, Chaetomium elatum, Ch. subspirilliferum, Cladosporium sphaerospermum, Fusarium oxysporum, F. tricinctum, Gymnoascus reesii, Parengyodontium album, Sporormiella minimoides, Stachybotrys chartarum,* and *Stemphylium vesicarium*) were shared between the culturable and DNA-based communities.

In the HH, NB, and NN soil profiles, 64, 41, and 77 species were respectively detected. At HH and NN, fungal communities contained a higher number of species in the biocrust layers while being more even in the 5–10 cm layer (Table 2b). At NB, the low species richness of the biocrust communities was probably due to the fact that the great majority of their taxa (92.1%) belonged to the lichen-forming fungi, which were excluded from the analysis.

The approximate calculations (due to the uncertainty in taxonomic affiliation of some comparatively abundant fungal phylotypes) also showed the prevalence of melanin-containing species in the crust communities, at least at NN and HH (63.7% and 72.4%, respectively), and high abundance of species with large many-celled spores, mostly from the genus *Alternaria* (60.3% and 60.5%). The decrease in abundance of these fungal groups, although less pronounced than in the culturable communities, was similarly revealed—both overall (3.2, 1.7, and 1.9-fold), and for species with multicellular spores (10.6, 1.1, and 2.6-fold in the NN, NB, and HH profiles, respectively). The increase in abundance of some light-colored species producing small one-celled spores—*Acremonium rutilum*, *Auxarthron alboluteum, Chrysosporium carmichaelii*, and *Gymnoascus reessii*, was also registered in the 5–10 cm layers of the studied profiles (Table 4).

Cluster analysis based on species relative abundances assembled the biocrust communities in one group, while the communities from 5 to 10 cm layers in the NN and HH profiles were clustered apart (Figure 5). The communities from the deeper soil layer of the NB profile grouped with the surface communities due to the comparatively high abundance of the same *Alternaria* sp.

## 4. Discussion

### 4.1. Effect of Altitude and Soil Depth on Fungal Communities

Using culture-dependent and culture-independent approaches, our study revealed diverse soil mycobiota accounting for 169 species. The results show that both altitude and soil depth significantly influenced the variations in composition, structure, and diversity of fungal communities. This conclusion was supported by univariate (ANOVA) and multivariate (AH clustering) analyses. Unfortunately, we could not apply the ANOVA test for the DNA-based communities because of the aforementioned uncertainty in taxonomic affiliation of some comparatively abundant fungal phylotypes.

Unexpectedly, fungal communities at the lowest study site, Nahal Nizzana (NN), were characterized by the highest species richness, probably due to the patchiness of habitats caused by high cover of pebbles and cobbles, which formed a mosaic of microsites with variable abiotic conditions [32,33]. The culturable communities at this site also contained the lowest and the highest proportions of melanin-containing fungi and light-colored species with small one-celled spores, respectively, at each depth of the studied profiles. At the same time, the absolute abundance—density of fungal isolates, was substantially lower at Nahal Nizzana, especially in the 5–10 cm soil layer, apparently due to the higher evaporation at this site (2700 mm in comparison to 2600 and 2500 mm at Nahal Boker (NB) and Har Harif (HH), respectively) and subsequently, the lower amounts of available water. Since all three sites benefit from similar amounts of rain, lower evaporation rates that stem from a decrease in temperature with elevation may indicate an improved water regime at the more elevated sites. As to the depth-wise distribution of isolate densities, the highest values found in the biocrust layers might be associated with higher content of organic matter in these layers [10].

The analyses showed that soil depth influenced more strongly fungal communities in comparison with altitude. Similarly, soil depth was the most powerful factor affecting the variation in microfungal communities at the Tabernas Desert, Spain, while the impact of plant canopy and slope orientation was much weaker [34]. The influence of soil depth was especially highly expressed in the communities’ structures. Expectedly, the uppermost biocrust layers at each site were dominated by melanin-containing species with many-celled, thick-walled, and large spores. Such species are well known stress-tolerant microorganisms resistant to solar and UV radiation, high temperature, desiccation, oligotrophic conditions, and chemical and radioactive pollution (e.g., [35] and references therein [36]), and their dominance is characteristic for almost all studied sites at the Negev Desert (e.g., [13]). Due to the prevalence of species with protective dark pigmentation and multicellular spore morphology, the biocrust communities from the sites at different altitudes were clustered into one group (Figure 3 and Figure 5). At the same time, in the layers of 1–5 cm and 5–10 cm, fungal communities were dominated by various light-colored species producing substantial amounts of small one-celled and thin-walled spores. The increase in abundance of such species might be associated with the penetration of massively produced very small fungal spores during water infiltration and their deposition mainly at the deeper soil layers, resembling the pattern revealed for the actinomycete spores in the high-recharge sites of the Pasco Basin at southeastern Washington state [37]. Probably at these depths, mesic fungi (like *Penicillium simplicissimum*) meet appropriate abiotic conditions (lower temperatures and higher organic matter content) for successful survival and competition with the stress-selected melanized fungi prevailing in the topsoil communities (like *Alternaria* spp.). Notably, the massive dominance of one or two species with small one-celled spores in the layers of 1–5 and 5–10 cm reduced species richness and evenness of their culturable communities, and this reduction was highly expressed in the soil profiles at NN where *Parengyodontium album* overwhelmingly prevailed in the deeper layers (Table 1 and Table 2).

### 4.2. Comparison of Fungal Communities Revealed by Culture-Dependent and Culture-Independent Approaches

Both culture-dependent and culture-independent methodologies revealed a similar general tendency in the composition of fungal communities, that is, the prevalence of melanin-containing species with many-celled, thick-walled, and large spores (mainly from the genus *Alternaria*) in the biocrust layers and the increase with depth in the proportion of light-colored species producing small one-celled spores. However, the composition of culturable communities and the DNA-based communities was substantially different, which resulted in only 13 species in common among a total of 169 species.

Expectedly, the diversity of culturable mycobiota in the studied area was remarkably lower than that of the DNA-based mycobiota—70 species from three soil layers and 113 species from two soil layers, respectively, which may be explained by several reasons.

It is a well-established fact that the soil dilution plate method used for the isolation of culturable fungal communities in our study has a set of limitations and biases (e.g., [38]). In the majority of soils, it does not allow the isolation of all culturable fungi following the necessity to dilute the soil sample in order to obtain the “readable” community on a Petri dish (in our case, we diluted the soil samples at ratios of 1:10 and 1:100). The soil dilution method can lead to the overestimation of heavily sporulating species and loss of those fungi unable to grow on culture media or requesting a specific isolation technique and special growth media. At the same time, the molecular approach, which does not have the above limitations, allows the detection of much more diverse mycobiota comprising a higher variety of phyla. The DNA-based analyses found representatives of zoospore-producing Chytridiomycota such as *Rhizophlyctis rosea*, which can survive in dry soil in the sporangium state [39]; it also includes the obligate arbuscular mycorrhizal species from the phylum Glomeromycota. The phylum Basidiomycota is represented by 13 species, among which is *Geminibasidium* sp. abundantly occurred at NN at the depth of 5–10 cm; species belonging to this soil genus are known for their heat resistance and xerotolerance [40]. Additionally, the molecular approach has detected representatives of a specific group of rock-inhabiting microcolonial fungi (e.g., [41]), such as *Knufia* spp. and *Vermiconia antarctica.* Moreover, the DNA-based mycobiota includes also species that can be considered exotic for desert soils; for example, the smut fungus *Urocystis tritici* and species producing relatively large aboveground fruit bodies, such as the cup fungi *Trichophaeopsis* sp., *Scutellinia* sp., and the cap fungi *Psathyrella arenosa* and *Agaricus gennadii*; these species were detected with comparatively high relative abundances in the study’s profiles (Table 4).

Like culture-dependent methodology, the molecular approach has its own set of limitations and biases, which can cause the aforementioned differences in the composition of fungal communities. The DNA-based fungal communities belong to so-called total communities including not only active and dormant members but also dead members, as opposed to the RNA-based communities containing exclusively active fungi [42]. At the same time, culturable communities include species able to grow on cultivation media, that is, fungi in an active or dormant state. Another problem is associated with the use of the ITS region as the universal fungal barcode locus [43], while for the detection of species from the family Aspergillaceae (the genera *Penicillium*, *Aspergillus*, and *Talaromyces*), this barcode may be problematic and other barcodes such as beta tubulin and calmodulin genes are recommended (e.g., [44]). Probably because of this fact, the DNA-based communities in our studied sites do not include the *Aspergillus* and *Talaromyces* species isolated from the biocrust and soil samples (Table 1). Moreover, former communities contain the unidentified *Penicillium* sp. in low relative abundances, while in the culturable communities, *P. simplicissimum* was highly abundant (even taking into account possible overestimation of this heavily sporulating species) in the deeper layers of the HH and NB soil profiles.

Unfortunately, nearly a fourth part of ASVs was identified only to a high-rank taxonomic level—order and phylum, which complicated the analysis of the DNA-based communities because the unidentified species were comparatively abundant in the studied profiles (Table 4). Certainly, some of these species may be novel for the science. Nevertheless, one of the major problems associated with the taxonomic affiliation of either specimen-based or environmental DNA sequences is that the sequence data currently exist for a number of named fungal species corresponding to about 30% of the morphologically known species, and not all of the existing sequence databases are well-curated [45]. It means that in some cases, morphological identification of fungal species may give more precise and reliable results than that provided by molecular identification.

## 5. Conclusions

A diverse mycobiota accounting for 169 species was found in the soil profiles along an altitudinal gradient in the Central Negev Desert. The impact of soil depth on the variations in fungal communities was stronger than that of altitude. Exposed to higher temperatures, radiation, and limited water supply, the communities from the uppermost biocrust layers were dominated by fungi possessing the protective features: dark pigmentation and multicellular spore morphology. At the same time, several species of light-colored fungi, producing a high amount of small thin-walled spores, prevailed in the deeper soil layers. The highest evaporation rate, decreasing water availability at the lowest altitude in Nahal Nizzana, possibly resulted in the decrease in a quantitative characteristic of culturable fungal communities—isolate density. The employment of both culture-dependent and culture-independent methodology allowed to reveal the diverse soil mycobiota from various taxonomic and ecological groups. Since both traditional and molecular approaches have their own biases and limitations, a combination of these approaches is preferable and can give a more comprehensive picture of fungal diversity in soils.

## Figures and Tables

**Figure 1 microorganisms-11-01761-f001:**
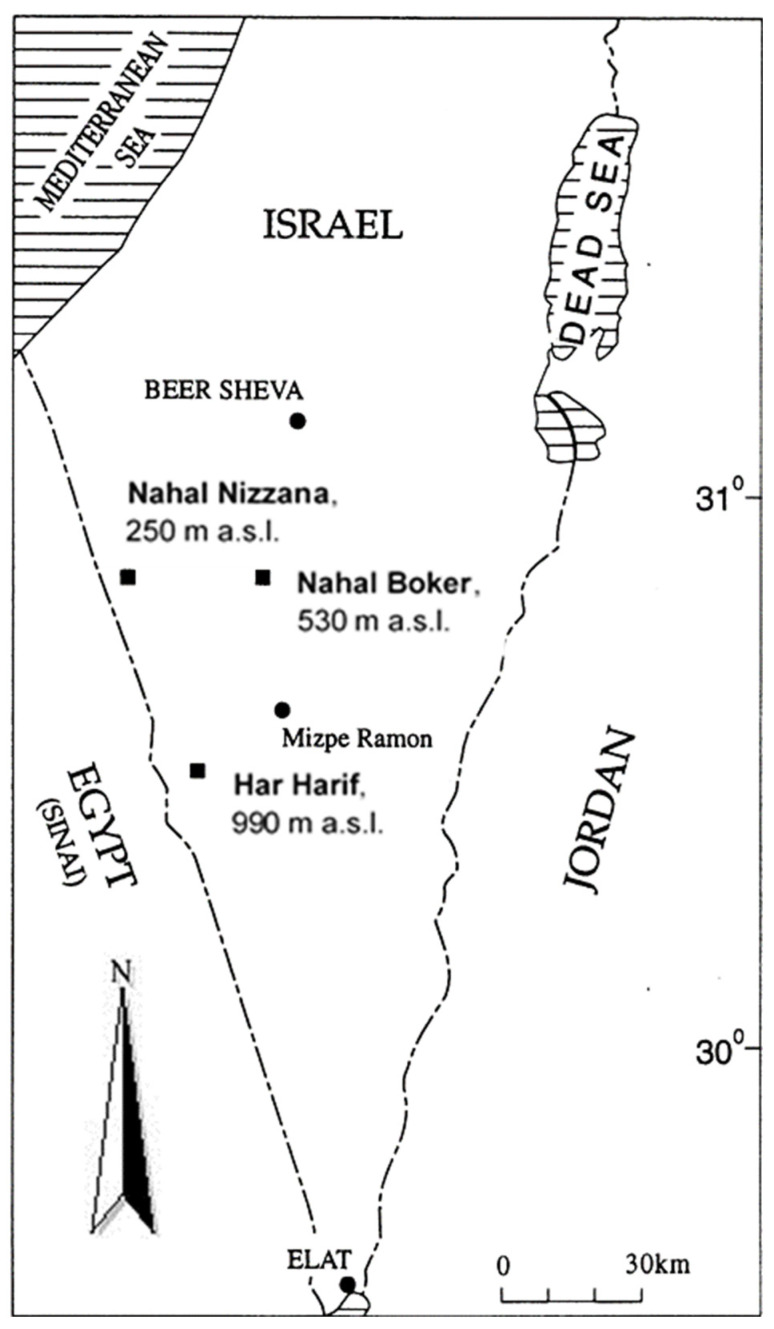
Map showing location of research sites (black squares) at the Central Negev Desert (adapted from [19]); black circles indicate cities.

**Figure 2 microorganisms-11-01761-f002:**
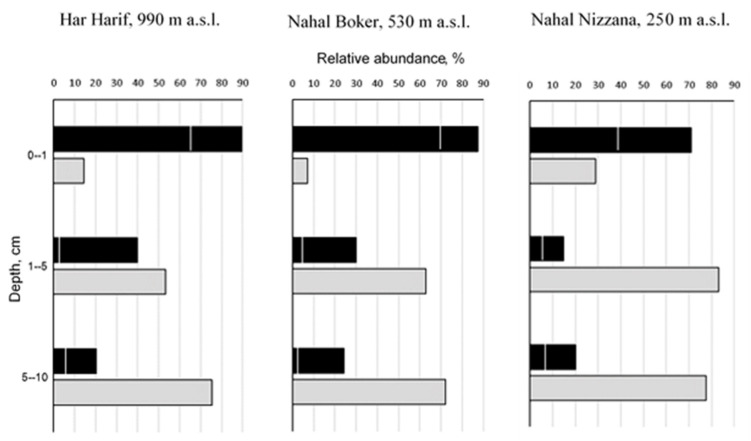
Dynamics of relative abundance of main groups of culturable fungal communities in soil profiles along altitudinal gradient at the Central Negev Desert: 
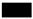
—melanin-containing spp.; 
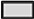
—light-colored spp. with small one-celled spores; area left from the line on the bars of melanin-containing spp. indicates contributions of species with large multicellular spores.

**Figure 3 microorganisms-11-01761-f003:**
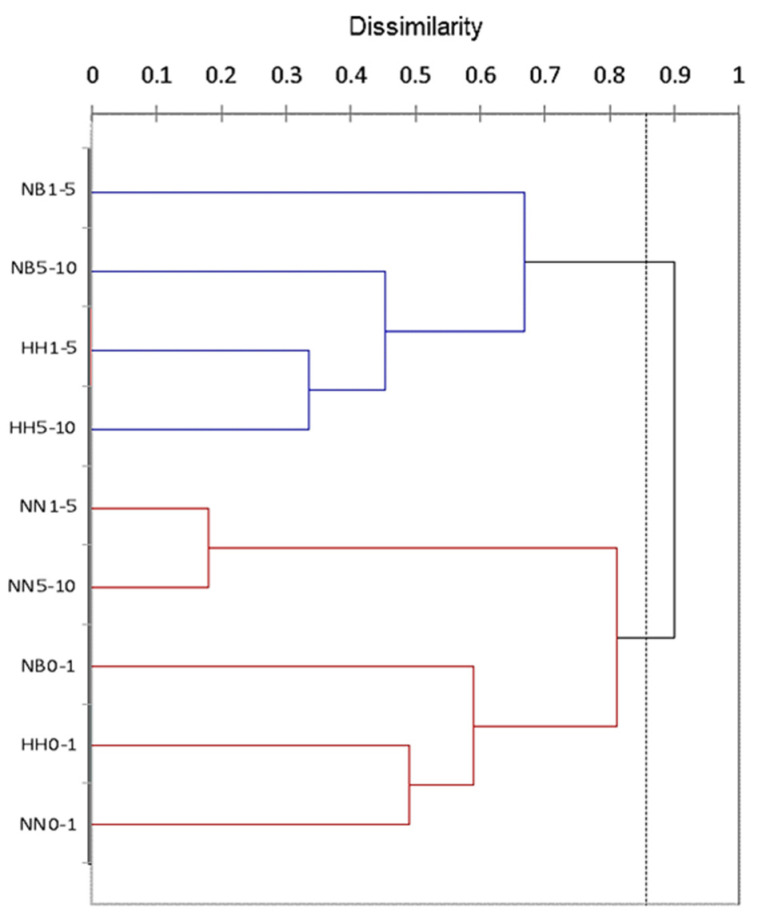
Clustering the culturable fungal communities from soil profiles along altitudinal gradient at the Central Negev Desert based on species relative abundance. Abbreviations: HH—Har Harif, 990 m.a.s.l.; NB—Nahal Boker, 530 m.a.s.l.; NN—Nahal Nizzana, 250 m.a.s.l. Numbers after abbreviations denote soil depth. The dotted line represents the automatic truncation, leading to two clusters.

**Figure 4 microorganisms-11-01761-f004:**
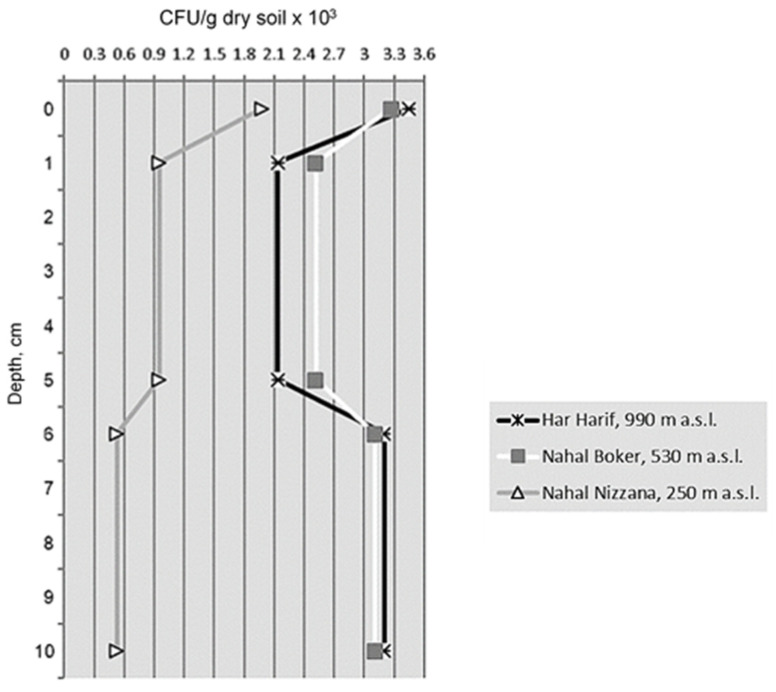
Dynamics of isolate density in soil profiles along altitudinal gradients at the Central Negev Desert.

**Figure 5 microorganisms-11-01761-f005:**
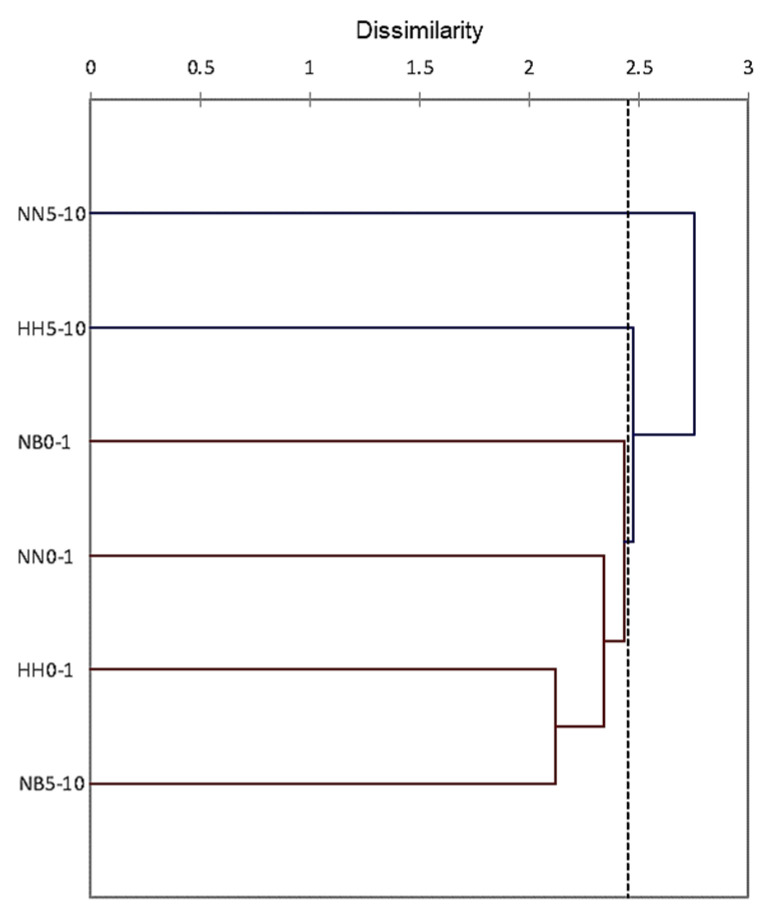
Clustering the DNA-based fungal communities from soil profiles along altitudinal gradient at the Central Negev Desert based on species relative abundance. Abbreviations: HH—Har Harif, 990 m.a.s.l.; NB—Nahal Boker, 530 m.a.s.l.; NN—Nahal Nizzana, 250 m.a.s.l. Numbers after abbreviations denote soil depth. The dotted line represents the automatic truncation, leading to one cluster and two separate classes.

**Table 1 microorganisms-11-01761-t001:** Fungi isolated in cultures from soil profiles along altitudinal gradient at the Central Negev Desert, with their relative abundance (%). Melanin-containing species are underlined; melanin-containing species with large multicellular spores are in bold; light-colored species with small (2–6 µ) one-celled spores are marked with an asterisk.

Species	HH 0–1 cm	HH 1–5	HH 5–10	NB 0–1	NB 1–5	NB 5–10	NN 0–1	NN 1–5	NN 5–10
Mortierellomycota									
** Mortierella humilis*	-	5.8	4.6	-	1.7	0.8	0.5	-	-
Ascomycota									
** Acremonium alternatum*	-	-	-	-	-	-	-	-	0.24
** * Acrospeira mirabilis * **	-	-	-	-	-	-	0.7	-	-
** * Alternaria alternata * **	20.6	-	0.3	4.7	0.15	-	14.6	1.3	2.9
** * A. atra * **	17.4	-	-	35.5	3.8	0.8	11.7	-	1.2
** * A. botrytis * **	-	-	-	-	-	-	-	-	0.24
** * A. chartarum * **	-	-	-	-	-	-	-	-	0.24
** * A. chlamydospora * **	-	-	-	-	-	1.6	0.25	-	0.5
** * A. chlamydosporigena * **	5.8	1.9	2.6	1.1	0.5	0.5	4.1	0.16	1
** * A. raphani * **	0.2	-	-	1.2	-	-	0.25	0.16	-
** Aphanocladium album*	1.7	3.0	0.8	0.3	0.5	-	-	1.7	0.5
* Arthrinium phaeospermum *	0.2	-	-	-	-	-	-	-	0.24
* Ascohyta medicaginicola *	0.9	-	-	-	-	-	-	-	-
** Aspergillus crustosum*	-	-	-	-	-	0.13	0.5	0.3	-
** A. fumigatus*	-	-	-	-	-	-	-	0.5	-
* A. niger *	0.4	-	-	0.3	-	-	0.25	-	-
** Auxarthron umbrinum*	-	-	-	-	0.5	-	-	0.16	-
* Boeremia exigua *	3.2	-	-	0.4	-	-	-	-	-
** * Camarosporium aequivocum * **	1.9	-	-	0.3	-	0.13	-	-	-
* Chaetomium elatum *	1.1	3.0	0.9	0.3	-	1.7	-	-	-
* Ch. globosum *	-	2.8	2.6	1.5	1.2	0.5	0.5	-	-
* Ch. hispanicum *	-	-	-	1.7	1.3	-	-	-	-
* Chaetomium piluliferum *	-	16.7	9.8	2.7	4.5	6.8	-	0.16	1
* Ch. strumarium *	-	-	-	-	-	-	0.25	-	-
* Ch. * cf. *subspirilliferum* ^1^	-	-	-	-	-	-	-	3.7	2.4
* Chaetomium * sp.	-	-	-	1.1	5.9	0.5	-	-	-
** Chrysosporium merdarium*	-	0.9	-	-	-	-	-	-	-
* Cladosporium cladosporioides *	8.8	0.23	0.5	4.7	0.7	-	26.8	0.5	5.9
* C. herbarum *	0.2	-	-	-	-	-	-	-	-
* C. sphaerospermum *	1.5	-	-	0.3	-	-	0.25	-	-
* Collariella bostrychodes *	-	8.8	0.15	-	-	-	-	-	-
*Cordyceps farinosa*	-	0.23	-	-	-	-	-	-	-
*Cylindrocarpon didymum*	-	-	0.3	-	-	-	-	-	-
** * Epicoccum nigrum * **	-	-	-	-	-	-	0.25	-	-
*Fusarium equiseti*	1.7	-	2.3	1.4	3.3	1.5	0.25	-	0.7
*F. oxysporum*	-	-	-	-	1.8	1.7	0.5	-	-
*F. sporotrichioides*	-	-	-	-	0.5	-	-	-	-
*F. tricinctum* ^2^	1.9	1.2	1.2	1.1	3.6	0.3	0.5	-	-
** Gymnoascus reessii*	-	-	5.2	5	42.8	20.6	1	-	-
*Helicodendron articulatum*	-	-	-	1.5	-	-	-	-	-
* Juxtiphoma eupyrena *	0.9	0.7	-	-	-	-	-	-	-
* Lasiodiplodia theobromae *	-	-	-	0.6	-	-	-	-	-
** Lecanicillium psalliotae*	-	-	-	0.3	-	-	-	-	-
* Massarina igniaria *	-	-	-	-	-	-	0.25	-	0.5
* Metulocladosporiella musae *	-	-	-	-	-	0.4	-	-	-
** Metarhizium marquandii*	1.5	-	-	-	-	-	-	-	-
* Microascus cirrosus *	-	-	-	-	12.2	-	-	-	-
* M. trigonosporus *	-	-	-	-	-	8.9	1.2	-	-
* Microsphaeropsis olivacea *	6.6	-	-	-	-	-	-	-	-
** * Neocamarosporium chichastianum * ** ^2^	0.2	0.23	-	2.1	0.7	-	-	-	0.7
* Neocucurbitaria cava *	0.2	0.5	-	-	-	-	-	1.8	0.7
* Nigrospora oryzae *	0.4	-	-	-	-	-	-	0.16	-
** Parengyodontium album*	3.2	4.4	3.2	0.8	0.8	0.4	23.7	87.2	75.4
** * Papulaspora pannosa * **	1.3	-	-	-	-	0.13	2.4	1.2	-
** Penicillium simplicissimum*	0.6	34.0	43.9	0.3	10.9	49.4	0.5	0.16	3.3
** Pseudogymnoascus pannorum*	-	3.7	5.7	-	1.7	1.3	-	-	-
* Pseudothielavia terricola *	-	-	-	-	-	-	0.25	-	-
** Sarocladium strictum*	-	0.7	-	-	-	0.3	-	-	-
** * Scolecobasidium humicola * **	-	-	-	-	-	-	-	0.16	-
* Sordaria fimicola *	0.2	-	-	-	-	-	-	-	-
** * Sporormiella australis * **	2.6	0.23	-	11.8	0.8	0.4	0.7	0.16	0.5
** * S. minimoides * **	-	-	-	1.8	-	-	0.25	-	-
* Stachybotrys chartarum *	1.5	-	0.15	1.2	-	0.13	2.9	-	0.7
** * Stemphylium botryosum * **	12.3	-	0.3	11.9	-	0.5	3.2	0.5	0.5
** * S. vesicarium * **	-	-	-	2.7	-	-	-	-	-
** Talaromyces variabilis*	-	1.9	13.7	-	-	-	-	-	-
* Trichocladium acropullum * ^2^	-	5.1	1.4	-	-	-	-	-	0.5
* Venustampulla parva *	0.2	-	-	-	-	-	-	-	-
Immature fruit bodies	0.2	-	-	-	0.15	-	-	-	-
Basidiomycota									
*Cutaneotrichosporon mucoides*	-	0.5	-	-	-	-	-	-	-

^1^ Identified by molecular analysis, 98% of maximal identity; ^2^ identified by molecular analysis, 99% of maximal identity.

**Table 2 microorganisms-11-01761-t002:** Diversity characteristics of fungal communities (*S*—number of species; *H*—Shannon index; *J*—evenness) in soil profiles along altitudinal gradient at the Central Negev Desert.

Locality		S			H			J		
	Depth	0–1 cm	1–5 cm	5–10 cm	0–1 cm	1–5 cm	5–10 cm	0–1 cm	1–5 cm	5–10 cm
a. Culturable communities										
Har Harif, 990 m.a.s.l.		32	23	21	2.54	2.18	2.01	0.73	0.69	0.63
Nahal Boker, 530 m.a.s.l.		32	23	25	2.38	2.12	1.71	0.69	0.68	0.53
Nahal Nizzana, 250 m.a.s.l.		31	18	23	2.14	0.67	1.15	0.62	0.23	0.37
b. DNA-based communities										
Har Harif		46	-	37	2.86	-	2.45	0.64	-	0.68
Nahal Boker		20	-	28	2.50	-	2.34	0.83	-	0.70
Nahal Nizzana		52	-	43	2.58	-	2.82	0.72	-	0.75

**Table 3 microorganisms-11-01761-t003:** Results of two-way unbalanced ANOVA analysis for the effect of altitude, soil depth, and interactions between them on different parameters of culturable fungal communities in soil profiles at the Central Negev Desert.

Parameter		R^2^	Adjusted R^2^	F^Pr>F^		
	Source			Altitude (DF = 2) ^1^	Soil Depth (DF = 2)	Altitude × Depth (DF = 4)
Species richness (S)		0.547	0.412	2.42 ^ns^	12.43 ****	0.72 ^ns^
Shannon index (H)		0.732	0.653	14.18 ****	16.97 ****	2.88 ^@^
Evenness (J)		0.654	0.551	12.02 ****	7.48 **	2.99 ^@^
Melanin-containing (MC) spp.		0.884	0.85	5.13 *	95.02 ****	1.26 ^ns^
MC spp. with large multicellular spores		0.918	0.893	2.74 ^ns^	137.01 ****	5.25 **
Light-colored spp. with small one-celled spores		0.886	0.852	8.36 ***	93.40 ****	1.58 ^ns^
Isolate density		0.458	0.298	7.53 **	1.61 ^ns^	1.14 ^ns^

^1^ DF—degree of freedom; ^@^ *p* ≤ 0.05; * *p* ≤ 0.01; ** *p* ≤ 0.005; *** *p* ≤ 0.001; **** *p* ≤ 0.0001; ns—non-significant.

**Table 4 microorganisms-11-01761-t004:** Fungi revealed by DNA extraction from soil profiles along altitudinal gradient at the Central Negev Desert, with their relative abundance (%). Melanin-containing species are underlined; melanin-containing species with large multicellular spores are in bold.

Species	HH 0–1 cm	HH 5–10 cm	NB 0–1 cm	NB 5–10 cm	NN 0–1 cm	NN 5–10 cm
Chytridiomycota						
*Powellomyces* sp.	0.3	-	-	-	-	-
*Rhizophlyctis rosea*	0.13	-	-	-	0.5	0.22
*Spizellomyces* sp.	-	-	-	-	-	0.44
Chytridiomycota*_*sp.	-	-	-	-	-	0.44
Glomeromycota						
*Claroideoglomus drummondii*	-	-	-	-	-	0.44
*Rhizophagus* sp.	-	-	-	-	-	0.11
Mortierellomycota						
*Mortierella alpina*	-	-	-	-	0.1	0.6
Ascomycota						
*Acremonium alternatum*	-	-	-	0.8	-	-
*A. rutilum*	7.8	-	-	-	-	-
*Acremonium* sp.	-	-	-	-	2.0	-
***Alternaria* sp.**	37.6	8.5	6.5	20.8	13.9	1.5
** *A. chlamydospora* **	4.8	-	1.3	4.3	0.1	0.7
** *A. chlamydosporigena* **	-	-	-	-	5.4	0.44
** *A. hungarica* **	1.0	-	-	-	2.1	-
** *A. obovoidea* **	1.8	-	-	0.33	0.1	-
** *A. oregonensis* **	0.9	-	-	0.44	0.4	-
* Ascobolus * sp * . *	0.5	-	-	-	-	-
*Auxarthron alboluteum*	-	-	-	-	-	16.8
*Auxarthron* sp.	-	-	-	-	-	1.1
* Botryotrichum spirotrichum *	0.7	5.5		2.4	0.3	2.1
* Botryotrichum * sp * . *	-	-	-	-	0.6	-
*Camarosporidiella* sp.	-	1.0	-	-	-	-
*Cephaliophora* sp.	-	1.3	-	0.11	-	-
* Chaetomium acropullum *	-	0.3	-	-	-	1.1
* Ch. elatum *	-	-	-	-	-	2.9
* Ch. iranianum *	-	-	-	-	0.3	-
* Ch. subspirilliferum *	-	-	-	-	0.1	-
* Chaetomium * sp * . *	-	3.8	-	-	-	1.2
*Chrysosporium carmichaelii*	-	21.1	-	1.3	-	-
* Cladosporium ossifragi *	0.3	-	-	-	1.0	-
* C. sphaerospermum *	0.6	0.14	1.3	0.11	0.4	-
* C. tenuissimum *	-	0.14	-	-	-	-
* Cladosporium * sp * . *	-	-	-	0.22	-	-
***Comoclathris* sp.**	-	0.6	1.3	-	0.4	-
*Darksidea* sp.	-	-	-	-	0.1	1.5
***Dematiopleospora* sp.**	0.13	0.14	-	-	0.1	-
* Foliophoma fallens *	-	-	-	-	-	-
*Fusarium oxysporum*	-	-	-	-	0.1	-
*F. redolens*	-	-	-	-	-	1.3
*F. tricinctum*	0.13	1.4	1.3	18.8	1.0	5.7
*Gymnoascus reessii*	-	0.14	-	5.3	0.1	6.3
* Knufia karalitana *	0.3	-	-	0.11	-	-
*Knufia* sp.	1.0	-	-	-	1.3	-
** *Libertasomyces myopori* **	0.13	1.0	-	-		-
***Libertasomyces* sp.**	-	-	-	-	0.7	-
*Mallochia reticulata*	-	-	-	-	0.2	-
*Monosporascus* sp.	-	1.3	-	-	0.2	3.4
*Monosporascus* sp.1	-	-	-	-	-	0.22
***Neocamarosporium* sp.**	4.3	2.4	6.5	3.7	0.2	-
* Neodevriesia * *sp.*	1.9	-	-	0.11	-	-
* Paraphaeosphaeria sporulosa *	0.13	1.4	1.3	-	-	-
*Parengyodontium album*	0.13	0.7	-	2.2	-	-
*Penicillium* sp.	-	0.6	-	2.2	-	0.6
* Phaeotheca triangularis *	0.5	-	1.3	-	-	-
** *Preussia flanaganii* **	1.6	-	-	-	0.8	0.22
** *P. terricola* **	-	-	-	-	-	2.4
***Preussia* sp.**	-	-	-	-	3.1	0.11
* Pseudocoleophoma * * calamagrostidis *	-	-	-	-	0.1	-
** *Pseudopithomyces * ** ** *chartarum* **	-	1.7	-	-	-	-
* Saxophila tyrrhenica *	0.13	-	-	0.11	-	-
* Schizothecium carpinicola *	-	-	-	-	0.7	-
* S. inaequale *	-	-	-	-	-	0.33
*Scutellinia* sp.	-	-	-	-	-	11.2
***Sigarispora* sp.**	-	-	-	-	0.5	0.11
** *Sporormiella megalospora* **	0.13	-	-	-	1.3	-
** *S. minimoides* **	0.13	-	-	-	0.1	-
* Stachybotrys chartarum *	0.13	-	-	-	-	-
***Stemphylium* sp.**	0.4	5.6	2.6	-	1.2	0.11
** *S. vesicarium* **	0.13	-	-	-	-	-
* Subramaniula * sp * . *	-	-	-	-	0.4	-
*Toxicocladosporium* sp.	-	-	-	-	0.1	-
*Trichophaeopsis* sp.	-	-	-	-	15.7	-
* Trimmatostroma salinum *	1.3	-	-	-	-	-
* Vermiconia antarctica *	4.4	1.8	6.5	0.11	0.1	-
* Verrucocladosporium dirinae *	-	-	-	-	0.1	-
* Zopfiella * sp * . *	-	-	-	-	-	0.33
Botryosphaeriales sp.	-	-	-	0.11	-	-
Capnodiales sp.	1.6	0.3	-	-	-	-
Capnodiales sp.1	-	0.4	7.8	-	-	-
Chaetothyriales sp.	-	-	-	-	-	2.3
Hypocreales sp.	-	-	25.9	-	-	-
**Pleosporales sp.**	1.7	2.5	3.9	2.0	26.0	0.11
**Pleosporales sp.1**	5.7	0.6	11.7	-	3.9	0.11
Xylariales sp.	-	0.14	-	1.5	-	7.2
Eurotiomycetes sp.	-	6.5	-	9.6	-	0.8
Ascomycota_ sp.1	0.13	-	-	20.2	3.8	-
Ascomycota_ sp.2	0.13	2.3	-	0.11	1.4	1.9
Ascomycota_ sp.3	4.7	-	1.3	-	-	-
Ascomycota_ sp.4	1.3	-	-	-	1.00	-
Ascomycota_ sp.5	-	-	-	-	-	2.0
Ascomycota_ sp.6	1.2	1.0	-	-	0.1	-
Ascomycota_ sp.7	-	1.3	-	-	-	-
Ascomycota_ sp.8	-	-	-	-	0.6	-
Ascomycota_ sp.9	0.5	-	-	-	-	-
Ascomycota_ sp.10	0.13	-	-	-	0.2	-
Ascomycota_ sp.11	-	-	3.9	-	-	-
Ascomycota_ sp.12	-	-	1.3	-	0.1	-
Ascomycota_ sp.13	2.3	-	-	-	-	-
Ascomycota_ sp.14	3.8	4.3	11.7	1.0	3.3	1.2
Ascomycota_ sp.15	-	0.14	1.3	-	-	-
Basidiomycota						
*Agaricus gennadii*	-	6.7	-	-	-	-
*A. velutipes*	2.3	-	-	-	-	-
*Ceratobasidium* sp.	-	-	-	-	-	0.22
*Coniophora* sp.	-	-	-	-	0.1	-
*Geminibasidium* sp.	-	-	-	1.9	-	18.5
*Lyomyces* sp.	-	-	-	-	2.7	-
*Malassezia restricta*	-	-	-	0.11	-	-
*Naganishia* sp.	-	-	-	-	0.7	-
*Psathyrella arenosa*	-	10.5	-	-	-	-
*Thanatephorus* sp.	0.7	-	-	-	-	-
*Urocystis tritici*	0.2	-	-	-	0.1	-
Basidiomycota_ sp.1	-	-	-	-	-	0.33
Basidiomycota_ sp.2	-	-	1.3	-	-	0.22

## Data Availability

No new data were created or analyzed in this study. Data sharing is not applicable to this article.

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
