# Peer review of "Altitudinal Gradient and Soil Depth as Sources of Variations in Fungal Communities Revealed by Culture-Dependent and Culture-Independent Methods in the Negev Desert, Israel"

_microorganisms, 2023, doi:10.3390/microorganisms11071761_

Round 1

Reviewer 1 Report

Dear autrors

Thank you for the opportunity to review yous manuscript and congratulation for this effort in to help in the understanding of desert fungi communities. I have only some small suggestions to the text, as folows: 

- In the abstract, put in a sentence the year in which the research was carried out (2020).

- In figure 1 indicate what the black circles (cities, villages?) are and indicate with an arrow where the north is.

- Line 112 – “36 soil samples  were collected and processed .. I could not understand…. 36 in each point ? Clarify please.

Lines 199 and 200; 209 – 210; 221 – scientific names not in italic.

Yours 

Author Response

Response to reviewers' comments

Reviewer 1

In the abstract, put in a sentence the year in which the research was carried out (2020).

       - The year of research has been added to the abstract (line 15)

In figure 1, indicate what the black circles (cities, villages?) are and indicate with an arrow where the north is.

  • In Figure 1, the indication of black circles has been explained, and the arrow pointed to the north has been inserted.

Line 112 – “36 soil samples were collected and processed .. I could not understand…. 36 in each point? Clarify please.

  • "Altogether from all soil profiles, 36 soil samples were collected and processed for the isolation of culturable fungal communities." – lines 115-117.

Lines 199 and 200; 209 – 210; 221 – scientific names not in italic.

  • corrected

Reviewer 2 Report

Dear Authors,
Your manuscript provides information on fungal biodiversity in desert soils. An interesting aspect is the comparison of two methodological approaches. I find the work interesting but have minor comments:

Line 93

Evaporation?

Line 108

Could you explain how the samples were collected? Was any equipment/tools used?

Line 145

Could you cite other papers in which these primers were used?

Lines 199-200, 209-210, 223,

Use italics for species name

Author Response

Response to reviewers' comments

Reviewer 2

Line 93 - Evaporation?

  • corrected

Line 108 - Could you explain how the samples were collected? Was any equipment/tools used?

  • The explanation has been added (lines 113-115)

Line 145 - Could you cite other papers in which these primers were used?

  • The study used the same primers has been cited (line 149)

Lines 199-200, 209-210, 223 - Use italics for species name

  • corrected